# Mental Illness and Youth-Onset Homelessness: A Retrospective Study among Adults Experiencing Homelessness

**DOI:** 10.3390/ijerph17228295

**Published:** 2020-11-10

**Authors:** Chisom N. Iwundu, Tzu-An Chen, Kirsteen Edereka-Great, Michael S. Businelle, Darla E. Kendzor, Lorraine R. Reitzel

**Affiliations:** 1Department of Rehabilitation and Health Services, College of Health and Public Service, University of North Texas, Denton, TX 76203, USA; kirsteen.ederekagreat@unt.edu; 2Department of Psychological, Health, and Learning Sciences, College of Education, University of Houston, Houston, TX 77204, USA; tchen3@central.uh.edu (T.-A.C.); lrreitzel@uh.edu (L.R.R.); 3HEALTH Research Institute, University of Houston, 4849 Calhoun Rd., Houston, TX 77204, USA; 4Oklahoma Tobacco Research Center, The University of Oklahoma Health Sciences Center, 655 Research, Parkway, Suite 400, Oklahoma City, OK 73104, USA; Michael-Businelle@ouhsc.edu (M.S.B.); Darla-Kendzor@ouhsc.edu (D.E.K.)

**Keywords:** mental illness, severe mental illness, youth-onset homelessness, homelessness onset, reason for homelessness, homeless youth versus adulthood

## Abstract

Financial challenges, social and material instability, familial problems, living conditions, structural issues, and mental health problems have been shown to contribute to youth homelessness. Based on the paucity of literature on mental illness as a reason for youth homelessness, the current study retrospectively evaluated the association between the timing of homelessness onset (youth versus adult) and mental illness as a reason for homelessness among homeless adults living in homeless shelters and/or receiving services from homeless-serving agencies. Homeless participants (*N* = 919; 67.3% men) were recruited within two independent studies from Dallas and Oklahoma. Covariate-adjusted logistic regressions were used to measure associations between homelessness onset and mental illness as a reason for current homelessness, history of specific mental illnesses, the historical presence of severe mental illness, and severe mental illness comorbidity. Overall, 29.5% of the sample reported youth-onset homelessness and 24.4% reported mental illness as the reason for current homelessness. Results indicated that mental illness as a reason for current homelessness (AOR = 1.62, 95% CI = 1.12–2.34), history of specific mental illnesses (Bipolar disorder–AOR = 1.75, 95% CI = 1.24–2.45, and Schizophrenia/schizoaffective disorder–AOR = 1.83, 95% CI = 1.22–2.74), history of severe mental illness (AOR = 1.48, 95% CI = 1.04–2.10), and severe mental illness comorbidities (AOR = 1.30, 95% CI: 1.11–1.52) were each associated with increased odds of youth-onset homelessness. A better understanding of these relationships could inform needs for early interventions and/or better prepare agencies that serve at-risk youth to address precursors to youth homelessness.

## 1. Introduction

Youths who experience homelessness bear a disproportionate burden of physical and mental health problems compared with their housed counterparts [1,2,3]. Homeless children and youth are described as individuals less than 25 years of age living in poverty and without a stable, reliable and adequate nighttime residence, and who may not have an alternative safe living environment [4,5]. Included in this group are accompanied and unaccompanied children (defined here as <18 years of age) and youth (defined here as 18–24 years of age) with and without a legal parent or guardian, respectively [6]. Unaccompanied youths experiencing homelessness are more likely to be unsheltered relative to other homeless groups [6]. Additionally, most unaccompanied individuals experiencing homelessness (89%) are between the ages of 18 and 24 (i.e., “youth”), and are particularly vulnerable to the detrimental and negative effects associated with homelessness [6,7]. In the United States (2019), about 31,000 individuals experienced homelessness as unaccompanied youth on a given single night [6].

Various issues including financial challenges, social and material instability, familial problems, living conditions, structural issues, etc., have been shown to contribute to youth homelessness [8]. A study exploring youths’ transition to homelessness showed that physical abuse and intense familial conflict were the most predominant factors that contributed to youth homelessness, and many homeless youths come from single-parent households [9]. In another study among homeless adults, a variety of circumstances were identified precursors to homelessness, such as unemployment leading to limited economic resources, addiction problems, and loss of support and relationships with family and friends [10]. Other predisposing factors to youth homelessness include maltreatment and violence, incarceration and transitional living, family dissolution and conflict, and exclusion from school because of frequent housing changes as a child [11].

Mental illness has also been reported as a reason for youth homelessness [1,12]. Among youth in general, the peak occurrence of most mental illnesses begins before the age of 25 [13,14]. In fact, about 75% of adults with a mental illness have experienced their first onset by 25 years of age [15,16]. According to the American Psychiatric Association, the average age of onset for depression is the mid-20 s [17]. Further, the median onset age for bipolar disorders is 24 [18], 22 for schizophrenia [13,19], and 20 for anxiety disorders [18,20]. Although post-traumatic stress disorder (PTSD) is not prevalent among youth, violence and trauma that precede youth onset homelessness, is highly reported among youth [7,21]. In addition, studies have shown that youths with mental illness are more likely to engage in a greater number of risk behaviors that may increase vulnerability to violent victimization, potentially leading to increased traumatic experiences that may consequently increase the likelihood of homelessness [22,23,24]. The occurrence of these mental illnesses could lead to increased morbidity and poor health outcomes. For example, exhibiting symptoms of bipolar disorder and depressive disorders at an earlier age is correlated with suicidal ideation and attempts, and psychiatric comorbidities [18,25]. Hence, not only do many serious mental illnesses occur during youth (ages 18–25), these mental illnesses tend to endure into middle age. Mental illness among youth could be the result of adverse childhood events such as neglect and abuse producing negative outcomes in adulthood, such as addiction and chronic diseases, and ultimately homelessness [26,27,28,29,30]. Therefore, there is a need for increased awareness and provision of mental health treatment programs to abate the issue of mental illness and to potentially prevent associated outcomes that may predispose individuals to youth-onset homelessness [31,32].

Moreover, the specific mental illness experienced by youth could be important in predicting youth-onset homelessness [1]. Some studies have found that homeless youths have greater prevalence of severe mental illness(es) such as major depression, bipolar disorder, and schizoaffective disorder, compared to homeless adults who predominantly report increased patterns of substance use as precursors of homelessness [3,29,33]. Anxiety disorders and post-traumatic stress disorder (PTSD) have also been reported to be common among homeless youths [34,35,36]. On the other hand, at least one study reported similar psychiatric comorbidities for homeless adults, except that homeless adults had a higher rate of psychosis than homeless youths [36]. Thus, the literature is somewhat mixed with regard to any patterns of mental illness that may predispose or correlate with youth-versus adult-onset homelessness.

Based on the paucity of literature on mental illness as a reason for youth homelessness, the purpose of this study was to evaluate the potential role of mental illness in homelessness onset in youth versus adulthood among homeless adults. Specifically, mental illness as a reason for the current homelessness episode and a history of specific mental illnesses, presence of severe mental illness, and severe mental illness comorbidity were examined regarding their correlations with homelessness onset in covariate-adjusted analyses. We hypothesized that several of these mental illness variables would be retrospectively associated with youth-onset homelessness among this group of middle-aged homeless adults. A better understanding of these relationships could inform needs for early interventions and/or better prepare agencies that serve at-risk youth to address precursors to youth homelessness.

## 2. Materials and Methods

### 2.1. Participants and Procedures

Adults experiencing homelessness were recruited within two independent studies on health and health risk factors. Participants for Study 1 were recruited in the summer of 2013 in Dallas, TX [37], and participants for Study 2 were recruited between July and August of 2016 from 6 homeless-serving agencies in Oklahoma City, OK [38]. Participants from both studies were recruited using flyers placed around the shelter areas and targeted settings. Inclusion criteria were: aged 18 or over, English-speaking, at least a 7th grade English literacy level as indicated by a score of >4 on the Rapid Estimate of Adult Literacy in Medicine-Short Form [39], and receiving services and/or shelter at the targeted agencies. Overall, 470 participants were screened in Study 1 and 648 were screened in Study 2, from which 76 (Study 1) and 38 (Study 2) individuals were ineligible for participation (for more information, see [37,40] and [38,41]). Overall, 394 (Study 1) and 610 (Study 2) participants enrolled. For the current study, we narrowed the samples to those who met the definition of being homeless [42]. All 394 enrollees in Study 1 resided at the shelter, whereas 525 in Study 2 met the HUD definition for current homelessness (i.e., self-identified as homeless and endorsed staying at a friend or family member’s house, homeless shelter, outside or on the street, hotel/motel, drug/alcohol treatment center, or other temporary location) [6]. Thus, the analyzable sample comprised 919 homeless adults.

All subjects gave their informed consent for inclusion before they participated in the study. The study was conducted in accordance with the Declaration of Helsinki, and the protocol was approved by the Institutional Review Boards of associated Institutions.

### 2.2. Measures

#### 2.2.1. Predictors

Mental illness as reason for current homelessness was assessed by asking participants “What are your reasons for current homelessness” with mental illness provided as an option coded as 0 = no and 1 = yes. Although reasons for current homelessness might differ from reasons for first homelessness, the chronicity of serious mental illnesses and their typical onset in youth/young adulthood is potentially concomitant with its perseverance as a precursor of the most recent homelessness episode. To ascertain details on the history of specific diagnoses, participants were asked to indicate whether they had a lifetime history of diagnosis of major depression, bipolar disorder, and/or schizophrenia/schizoaffective disorder, respectively (0 = no and 1 = yes). Participants from Study 2 were additionally asked to indicate if they had a lifetime history of post-traumatic stress disorder and/or other anxiety disorders (0 = no and 1 = yes). Additionally, 2 investigator-generated variables were constructed. The first was the presence of severe mental illness by history, which included major depression, bipolar disorder, and schizophrenia/schizoaffective disorder [43,44,45] whereby endorsement of any of the three yielded a 1 = historical presence of severe mental illness versus endorsement of none, which yielded 0 = no historical presence of severe mental illness. The second investigator-generated variable was indicative of severe mental illness comorbidity by history, and consisted of the number of severe mental illness diagnoses as assessed above summed with a potential range of 0–3.

#### 2.2.2. Outcome

Homelessness onset was assessed by asking “How old were you when you first became homeless?” with responses dichotomized as youth-onset homelessness (≤24 years) and adult-onset homelessness (>24 years). These age categorizations are consistent with previous studies examining youth versus adult-onset homelessness [29,46].

#### 2.2.3. Covariates

Participants were asked about their education (<GED/high school vs. ≥GED/high school), race (white, black, vs. other), sex (male vs. female), insurance status (insured vs. uninsured), employment status (employed vs. unemployed), marital status (married vs. unmarried), and annual income (≤$5000 vs. >$5000). They were also asked to report their lifetime (“What is the total amount of time you have been homeless in your lifetime?”) and current (“How long ago did the current period of homelessness begin?”) period of homelessness, and the number of discrete homelessness episodes. Data origin (Study 1 versus Study 2) was also included as a covariate.

### 2.3. Analytic Plan

The descriptive statistics of sample characteristics and study variables were assessed, and the comparisons between homelessness onset (youth versus adult) were examined using Mann–Whitney test or chi-square test for continuous and categorical variables, respectively. Covariate-adjusted logistic regressions were used to measure associations of homelessness onset with mental illness as a reason for current homelessness, as well as with a history of specific mental illnesses, the historical presence of severe mental illness, and severe mental illness comorbidity by history. All analyses used two-tailed tests of significance with a statistical significance level designated at *p* < 0.05. All analyses were conducted using SAS version 9.4 [47].

## 3. Results

### 3.1. Participants’ Characteristics

The sample (*N* = 919) comprised 67.25% males (*N* = 618). Overall, 54.95% of all participants (*N* = 505) self-identified as being of a racial/ethnic minority group (i.e., 38.74% Black or African American, and 16.21% other), and 24.37% (*N* = 224) reported mental illness as reason for current homelessness. See Table 1 for participants’ characteristics overall and by homelessness onset. Relative to those who reported adult-onset homelessness, participants who reported youth-onset homelessness were significantly more likely to have more homeless episodes (3, IQR: 2–5 vs. 2, IQR: 1–3), report mental illness as reason for current homelessness (31.37% vs. 21.45%), report the presence of severe mental illness by history (70.11% vs. 61.27%), and report severe mental illness comorbidity by history (1, IQR: 0–2 vs. 1, IQR: 0–2). Those reporting youth-onset homelessness were more likely to have a history of diagnosis with major depression (66.42% vs. 58.64%), bipolar disorder (42.07% vs. 28.24%), and schizophrenia/schizoaffective disorder (29.15% vs. 15.74%) relative to participants who indicated adult-onset homelessness. For Study 2, which also queried PTSD and other anxiety disorders, there were no significant differences in the prevalence by history of either, relative to homelessness onset.

### 3.2. Main Analyses

Table 2 reports the results of covariate-adjusted logistic regressions for associations between homelessness onset and mental illness as reason for current homelessness, a history of specific mental illnesses, the historical presence of severe mental illness, and severe mental illness comorbidity by history.

#### 3.2.1. Relation between Mental Illness as Reason for Current Homelessness and Homelessness Onset

The odds of changing from reporting no mental illness as reason for current homelessness to reporting mental illness as reason for current homelessness was 1.617 (95% CI: 1.118–2.339) for youth-onset homelessness versus adult-onset homelessness. That is, the expected likelihood of having the first homeless episode during youth versus adulthood was significantly greater for those who reported mental illness as reason for their current homelessness. Additionally, being other race (vs white; *p* = 0.0202) and having fewer months of homelessness in the current homelessness episode (*p* = 0.0178), more homelessness episodes over the lifetime (*p* < 0.0001), and being from the Dallas study site (*p* = 0.0303) were associated with greater odds of youth- versus adult-onset homelessness in this analysis.

#### 3.2.2. Relation between History of Specific Mental Illnesses and Homelessness Onset

Participants who were diagnosed with bipolar disorder and schizophrenia/schizoaffective disorder had 1.746 (95% CI: 1.242–2.453) and 1.831 (95% CI: 1.223–2.739) times greater odds of experiencing youth- versus adult-onset homelessness than those with no bipolar disorder or schizophrenia/schizoaffective disorder diagnosis, respectively. Moreover, being other race (vs white; *p* = 0.0224) and having fewer months of homelessness in the current homelessness episode (*p* = 0.0182), and more homelessness episodes over the lifetime (*p* < 0.0001) were associated with greater odds of youth- versus adult-onset homelessness in analysis with bipolar disorder. Fewer months of homelessness in the current homelessness episode (*p* = 0.0108), more homelessness episodes over the lifetime (*p* < 0.0001), and being from the Dallas study site (*p* = 0.0278) were associated with greater odds of youth- versus adult-onset homelessness in analysis with schizophrenia/schizoaffective disorder. However, the association between having major depression and homelessness onset was not significant (*p* = 0.1040). Having fewer months of homelessness in the current homelessness episode (*p* = 0.0172) and more homelessness episodes over the lifetime (*p* < 0.0001), and being from the Dallas study site (*p* = 0.0264), however, were associated with greater odds of youth- versus adult-onset homelessness in this analysis. Lastly, the association between having anxiety disorder, PTSD and homelessness onset on a subsample (Study 2) was not significant in covariate-adjusted analyses (*p* = 0.4841 and *p* = 0.6789, respectively). However, having more homelessness episodes over the lifetime (*p* < 0.0001) was associated with greater odds of youth- versus adult-onset homelessness in analyses with other anxiety disorders and PTSD.

#### 3.2.3. Relation between Historical Presence of Severe Mental Illness and Homelessness Onset

Adjusted odds of youth-onset homelessness to adult-onset homelessness was significantly greater among those who had a historical presence of severe mental health diagnoses compared with those who did not, with an odds ratio of 1.478 (95% CI: 1.041–2.098). Additionally, being other race (vs white; *p* = 0.0213) and having fewer months of homelessness in the current homelessness episode (*p* = 0.0153), more homelessness episodes over the lifetime (*p* < 0.0001), and being from the Dallas study site (*p* = 0.0275) were associated with greater odds of youth- versus adult-onset homelessness.

#### 3.2.4. Relation between Severe Mental Illness Comorbidity by History and Homelessness Onset

The adjusted logistic regression showed that each additional increase in a severe mental illness diagnosis was associated with 30% increased odds of youth- versus adult-onset homelessness (AOR: 1.301, 95% CI: 1.113–1.521). Additionally, being other race (vs white; *p* = 0.0207), and having fewer months of homelessness in the current homelessness episode (*p* = 0.0122), more homelessness episodes over the lifetime (*p* < 0.0001), and being from the Dallas study site (*p* = 0.0486) were associated with greater odds of youth- versus adult-onset homelessness in this analysis.

## 4. Discussion

This study explored the association between mental illness and homelessness onset in youth versus adulthood. As hypothesized, our findings support that mental illness—in particular, severe mental illness and its comorbidities as driven by bipolar disorder and schizophrenia/schizoaffective disorder- was associated with youth-onset homelessness. The results are consistent with previous studies [48,49,50,51] and expands upon prior literature by assessing relations with history of severe mental illness and its comorbidities, and examined these associations retrospectively among a large sample of homeless adults. Given that both bipolar and schizophrenia/schizoaffective disorder have been shown to typically emerge during youth (~22–24 years of age) [13,18,19,52], results linking a history of these disorders with the reason for current adult homelessness and youth-onset homelessness suggest that youths with mental illness may face compounding stressors and barriers in the access and delivery of mental health care which could result in youth-onset homelessness that perpetuates homelessness as an adult [5,33].

Significant associations between youth-onset homelessness and major depression were not supported in adjusted analyses, suggesting that the significant associations found between the historical presence of severe mental illness and youth-onset homelessness may be driven by bipolar and schizophrenia/schizoaffective disorder. Future work should further investigate what differentiates major depression from these other severe mental health disorders with regard to the potential link with youth-onset homelessness. In this adult sample, major depression was among the most prevalent of the mental health disorders investigated, reported by 61% of the sample. Thus, its presence may be so common among individuals experiencing homelessness as adults that it fails to distinguish the timing of homelessness onset [1]. Major depression may also be distinguished from bipolar and schizophrenia/schizoaffective disorder via a lack of prominent psychotic symptoms [53,54]. Medications used to treat psychotic disorders, relative to those used to treat major depression, may have unique side effects that prohibit ideal adherence [55,56], thereby contributing to the continued display of symptoms and the social and practical consequences of such, which may include youth-onset homelessness. Future studies should further examine these possibilities prospectively, including the potential contribution of medication non-adherence (by class) to homelessness onset.

Despite the fact that major depression was not an independent predictor of youth-onset homelessness, it is notable that mental illness comorbidity, which could include major depression, was associated with a greater likelihood of experiencing homelessness as a youth. Thus, the presence of major depression in conjunction with other disorders in young adulthood might confer additional risk for homelessness in youth relative to the presence of a singular severe mental illness. Research has shown a greater likelihood of developing a second mental illness after a first mental disorder diagnosis [57,58]. Thus, not only would mental illness diagnosis increase the likelihood of youth-onset homelessness, but once homeless, the likelihood of severe mental illness comorbidity may also increase. These results highlight the need for interventions focused on preventing comorbid mental health problems among at-risk youth, which could further prevent youth-onset homelessness, and the combined effect of dealing with both homelessness and mental illness comorbidities. Finally, prolonged homelessness compounded by myriad mental health problems could affect youths’ ability to engage in normative experiences in school, social and community circles, and in entering into the workforce, which could consequently contribute to chronic homelessness [59,60].

Similar to findings with depression, neither anxiety nor PTSD by history was associated with youth-onset homelessness among the proportion of the sample for which this information was available. Although these disorders may also occur in young adulthood, they appeared to contribute less to early homelessness than some of their more “severe” diagnostic counterparts. More research is needed to understand the reasons behind these results and their potential coalescence with those associated with null results for major depression as a distinguisher of youth- versus adult-onset homelessness. It is important to note that the lack of findings related to these diagnoses and youth-onset homelessness does not negate that traumatic experiences and resulting symptomatology (not rising to a diagnostic level) may be prevalent–and treatable–among youth at risk for homelessness. Future studies might expand to better capture non-diagnostic symptomatology and/or history of traumatic events and outcomes to better understand this possibility. Future work on this topic should also examine the relative effect, and the timing of the effects, of various disorders on social and workplace functioning.

Overall, these results highlight the need for the provision of mental health services for youth at risk for mental illness and homelessness. Particularly important in providing these services is the ability to recognize and identify those youth at risk of mental illness via various avenues such as schools, churches, workplaces, youths in juvenile detention centers, foster care, and residential treatment centers. Identification of at-risk youth for mental illness involves a recognition of warning signs that can be precursors to mental illness (e.g., youth experience of abuse and domestic violence, suicidal thoughts, motor vehicle accidents, sexually transmitted infection diagnoses, crime, and physical fights) and initial manifestations of severe mental illness (e.g., reductions in and/or non-developmentally appropriate psychological, social, and occupational functioning) by those providing services to youth [61]. An example intervention at the school level for identification of at-risk youth may involve the provision of services focused on social and emotional health, and screening tools for the identification of psychological and behavioral problems. It is important to identify youth at-risk for mental illness as they are a vulnerable group at higher risk of negative health outcomes [62]. It is also important to focus attention on at-risk youth groups, such as youth involved in foster care, to prevent homelessness and its negative effects on youth [4]. Identifying youths and providing them with appropriate care may have an impact on future homelessness, and could also lead to improved quality of care provided to homeless youth. The use of predictive algorithms and models could be particularly helpful in identifying these at-risk youth and could assist with the development of plans to change the trajectory of occurrence [62]. Further understanding mental illnesses faced by youth is instrumental to the development of successful targeted interventions. Future work should examine whether intervening to prevent and/or appropriately treat severe mental illness in young adulthood helps to reduce the likelihood of youth-onset homelessness as well as the chronicity of homelessness through adulthood. It is important to note that lack of insurance, however, could pose as a barrier to the accessibility of mental health and other support services, whereby access is limited for youth who are uninsured [63].

There are some strengths and limitations of our study. A major strength of the study is the large sample size derived by a merge of two homeless samples recruited from multiple sites across two states. Though limited samples (only Study 2) were used in specific analyses, a majority of the reported meaningful results were reported with the full sample size. There are also several limitations. The first is the cross-sectional nature of the study, which limits causal inferences. Thus, longitudinal studies will be valuable to further elucidate the measured associations. Another limitation concerns the use of self-reported measures, which could result in reporting bias. As such, future studies could incorporate the use of more objective data such as a medical record or a diagnostic interview. Our data were collected in the West South-Central region of the United States and might not be generalizable to other homeless samples in other locations. Despite these limitations, our samples (from Dallas and Oklahoma City) were representative of the homeless individuals in those cities [64,65]. Mental illness rates might also not be accurately reflective of the actual rate in our sample, as vulnerable groups are frequently underdiagnosed [66]. In addition, self-reporting a history of mental illness might also require participants to know and understand their diagnoses; however, effective doctor-patient communication might be lacking in some cases [67,68]. Additionally, a limitation of this work is that mental illness was assessed as the reason for *current* homelessness, which may be different than the reason for the *first* episode of homelessness. However, we adjusted for number of episodes of homelessness to account for this limitation. Additionally, we did not assess all possible historical mental illnesses, which may mean that other mental illnesses are important in differentiating youth- versus adult-onset homelessness. However, the assessed mental illnesses in this study have been shown in literature to be the most prevalent among individuals who are homeless [29,44]. We also do not have information on the timing of mental health diagnoses; thus, we do not truly know if mental illness or mental illness comorbidities preceded first homelessness. However, some of the diagnoses assessed are known to first emerge prior to age 25 (e.g., schizophrenia) [52].

## 5. Conclusions

The current study adds to a limited literature on mental health disorders as a distinguishing factor in homelessness onset. Results suggest that a history of specific mental illnesses–namely bipolar disorder and schizophrenia/schizoaffective disorder–as well as the historical presence of severe mental illness and severe mental illness comorbidities may be associated with reason for adult homelessness as well as youth vs. adult-onset homelessness. Results from this study highlight the need for innovative programs to address mental health issues among youth who are either at risk for mental illness, exhibiting early signs of mental illness, or at high risk for homelessness. Overall, a better understanding of these relationships could inform needs for early interventions and/or better prepare agencies that serve at-risk youth to address precursors to youth homelessness.

## Figures and Tables

**Table 1 ijerph-17-08295-t001:** Participants’ characteristics by homelessness onset (*N* = 919).

Variable	Homelessness Onset			
	Total Sample *N* = 919 *N* (%)/M [IQR]	Youth-Onset 271 (29.49%) *N* (%)/M [IQR]	Adult-Onset 648 (70.51%) *N* (%)/M [IQR]	*p*-Value
**Sociodemographic variables**				
*Education*				0.0732
<GED/high school	622 (67.68)	195 (71.96)	427 (65.90)	
≥GED/high school	297 (32.32)	76 (28.04)	221 (34.10)	
*Race*				0.0221
White	414 (45.05)	114 (42.07)	300 (46.30)	
Black	356 (38.74)	99 (36.53)	257 (39.66)	
Other	149 (16.21)	58 (21.40)	91 (14.04)	
*Sex*				0.9706
Male	618 (67.25)	182 (67.16)	436 (67.28)	
Female	301 (32.75)	89 (32.84)	212 (32.72)	
*Health insurance*				0.0972
Uninsured	672 (73.12)	188 (69.37)	484 (74.69)	
Insured	247 (26.88)	83 (30.63)	164 (25.31)	
*Employment status*				0.4673
Unemployed	821 (89.34)	239 (88.19)	582 (89.81)	
Employed	98 (10.66)	32 (11.81)	66 (10.19)	
*Marital status*				0.5321
Unmarried	836 (90.97)	249 (91.88)	587 (90.59)	
Married	83 (9.03)	22 (8.12)	61 (9.41)	
*Income*				0.5218
≤$5000	646 (72.75)	186 (71.26)	460 (73.37)	
>$5000	242 (27.25)	75 (28.74)	167 (26.63)	
*Site*				
Dallas, Texas	394 (42.87)	124 (45.76)	270 (41.67)	0.2533
Oklahoma City, Oklahoma	525 (57.13)	147 (54.24)	378 (58.33)	
*Lifetime homeless months*	30 [10.17–52.9]	37 [15.43–60]	25 [7.9–48]	<0.0001
*Current months of homelessness*	4 [1–10]	4 [1–11]	3.87 [0.7–9.7]	0.2967
*Number of homeless episodes*	2 [1–4]	3 [2–5]	2 [1–3]	<0.0001
**Reasons for homelessness variables**				
*Mental illness (as reason for current homelessness)*				0.0014
No	695 (75.63)	186 (68.63)	509 (78.55)	
Yes	224 (24.37)	85 (31.37)	139 (21.45)	
*History of specific mental illness*				
Major depression				
No	359 (39.06)	91 (35.58)	268 (41.36)	0.0275
Yes	560 (60.94)	180 (66.42)	380 (58.64)	
Bipolar disorder				
No	622 (67.68)	157 (57.93)	465 (71.76)	<0.0001
Yes	297 (32.32)	114 (42.07)	183 (28.24)	
Schizophrenia/schizoaffective disorder				
No	738 (80.30)	192 (70.85)	546 (84.26)	<0.0001
Yes	181 (19.70)	79 (29.15)	102 (15.74)	
PTSD *				
No	363 (69.14)	96 (65.31)	267 (70.63)	0.2353
Yes	162 (30.86)	51 (34.69)	111 (29.37)	
Other Anxiety disorder *				
No	321 (61.14)	84 (57.14)	237 (62.70)	0.241
Yes	204 (38.86)	63 (42.86)	141 (37.30)	
*Historical presence of severe mental illness*				0.0109
No	332 (36.13)	81 (29.89)	251 (38.73)	
Yes	587 (63.87)	190 (70.11)	397 (61.27)	
*Severe mental illness comorbidity by history*	1 [0–2]	1 [0–2]	1 [0–2]	<0.0001

Note. * Analyses conducted using sample from Study 2 (*N* = 525); PTSD = Post-Traumatic Stress Disorder; M [IQR] = Median [Interquartile range].

**Table 2 ijerph-17-08295-t002:** Logistic regression of homelessness onset and association with mental illness (*N* = 919).

Variables	Youth-Onset Homelessness OR (95% CI) (Ref Group = Adult-Onset)	*p*-Value
*Mental illness (as reason for current homelessness)*		
Yes (Ref: No)	1.617 (1.118, 2.339)	0.0107
*History of specific mental illness*		
Depression Yes (Ref: No)	1.326 (0.944, 1.563)	0.1040
Bipolar disorder Yes (Ref: No)	1.746 (1.242, 2.453)	0.0013
Schizophrenia/schizoaffective disorder Yes (Ref: No)	1.831 (1.223, 2.739)	0.0033
PTSD * Yes (Ref: No)	1.103 (0.693, 1.758)	0.6789
Other anxiety disorder * Yes (Ref: No)	1.167 (0.758, 1.796)	0.4841
*Historical presence of severe mental illness*		
Yes (Ref: No)	1.478 (1.041, 2.098)	0.0288
*Severe mental illness comorbidity by history*	1.301 (1.113, 1.521)	0.0010

Note. Covariates include education, race, sex, insurance status, employment status, marital status, income, lifetime homeless months, current months of homeless, number of homeless episodes, and study site; OR: odds ratio; CI: confidence interval; * Analyses conducted using sample from Study 2 (*N* = 525); PTSD = Post-Traumatic Stress Disorder.

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
