# Peer review of "Mental Illness and Youth-Onset Homelessness: A Retrospective Study among Adults Experiencing Homelessness"

_ijerph, 2020, doi:10.3390/ijerph17228295_

Round 1

Reviewer 1 Report

This paper examines the role of mental illness among those experiencing homelessness to see whether mental illness is linked to homelessness onset in youth versus adulthood. The study combines data from two different locations in the US and examines a very important issue among this population in which little research exists. The paper is very well-written and described. The only change i would like to see is interpretation of the covariates in Table 2. In other words, it was unclear if any of them were significant. Please add a note to the paper about their significance (or lack there of). Well done!

Author Response

Response 1: Thank you for your comments regarding our manuscript. We have mentioned this important point in the revised manuscript (lines 191-195), as below:

“Additionally, being other race (vs white; p=0.0202) and having fewer months of homelessness in the current homelessness episode (p=0.0178), more homelessness episodes over the lifetime (p<0.0001), and being from the Dallas study site (p=0.0303) were associated with greater odds of youth- versus adult-onset homelessness in this analysis”

And lines (197-215)

Participants who were diagnosed with bipolar disorder and schizophrenia/schizoaffective disorder had 1.746 (95% CI: 1.242 – 2.453) and 1.831 (95% CI: 1.223 – 2.739) times greater odds of experiencing youth- versus adult-onset homelessness than those with no bipolar disorder or schizophrenia/schizoaffective disorder diagnosis, respectively. Moreover, being other race (vs white; p=0.0224) and having fewer months of homelessness in the current homelessness episode (p=0.0182), and more homelessness episodes over the lifetime (p<0.0001) were associated with greater odds of youth- versus adult-onset homelessness in analysis with bipolar disorder. Fewer months of homelessness in the current homelessness episode (p=0.0108), more homelessness episodes over the lifetime (p<0.0001), and being from the Dallas study site (p=0.0278) were associated with greater odds of youth- versus adult-onset homelessness in analysis with schizophrenia/schizoaffective disorder. However, the association between having major depression and homelessness onset was not significant (p=0.1040). Having fewer months of homelessness in the current homelessness episode (p=0.0172) and more homelessness episodes over the lifetime (p<0.0001), and being from the Dallas study site (p=0.0264), however, were associated with greater odds of youth- versus adult-onset homelessness in this analysis. Lastly, the association between having anxiety disorder, PTSD and homelessness onset on a subsample (Study 2) was not significant in covariate-adjusted analyses (p=0.4841 and p=0.6789, respectively). However, having more homelessness episodes over the lifetime (p<0.0001) was associated with greater odds of youth- versus adult-onset homelessness in analyses with other anxiety disorders and PTSD.

And lines (219-222)

“Additionally, being other race (vs white; p=0.0213) and having fewer months of homelessness in the current homelessness episode (p=0.0153), more homelessness episodes over the lifetime (p<0.0001), and being from the Dallas study site (p=0.0275) were associated with greater odds of youth- versus adult-onset homelessness.”

And lines (226-229)

“Additionally, being other race (vs white; p=0.0207), and having fewer months of homelessness in the current homelessness episode (p=0.0122), more homelessness episodes over the lifetime (p<0.0001), and being from the Dallas study site (p=0.0486) were associated with greater odds of youth- versus adult-onset homelessness in this analysis.”

Reviewer 2 Report

The reviewer found this to be a very readable and thorough report. Much care has gone into the writing. Two minor corrections are recommended.

In line 66, "violence, trauma, etc. which precedes" should be "precede." Perhaps the entire line could be rewritten with the "etc." removed. In line 256 you again write that PTSD is not prevalent, which leaves the question of trauma (not reaching the level of PTSD) dangling, unfortunate as trauma (and trauma-related depression) might be the most treatable of all the presenting problems indicated in this group. 

Line 61 is about homeless youth. The rest of the paragraph is about youth in general. Please clarify this so the reader knows when you speak of homeless youth versus youth overall. Lines 80 and 81 also deserve a re-write, as do lines 90 to 92.

Thank you for this important paper.

Author Response

Response to Reviewer 2 Comments

The reviewer found this to be a very readable and thorough report. Much care has gone into the writing. Two minor corrections are recommended.

Point 1: In line 66, "violence, trauma, etc. which precedes" should be "precede." Perhaps the entire line could be rewritten with the "etc." removed.

Response 1: Thank you for your comments regarding our manuscript. We have changed the word “precedes” to “precede” (Line 68), and also removed “etc.” from the sentence (Line 67).

Point 2: In line 256 you again write that PTSD is not prevalent, which leaves the question of trauma (not reaching the level of PTSD) dangling, unfortunate as trauma (and trauma-related depression) might be the most treatable of all the presenting problems indicated in this group.

Response 2: Thank you for your important comments. We have mentioned this important point in the revised manuscript (lines 277-289), as below:

Similar to findings with depression, neither anxiety nor PTSD by history was associated with youth-onset homelessness among the proportion of the sample for which this information was available. Although these disorders may also occur in young adulthood, they appeared to contribute less to early homelessness than some of their more “severe” diagnostic counterparts. More research is needed to understand the reasons behind these results and their potential coalescence with those associated with null results for major depression as a distinguisher of youth- versus adult-onset homelessness. It is important to note that the lack of findings related to these diagnoses and youth-onset homelessness does not negate that traumatic experiences and resulting symptomatology (not rising to a diagnostic level) may be prevalent – and treatable – among youth at risk for homelessness. Future studies might expand to better capture non-diagnostic symptomatology and/or history of traumatic events and outcomes to better understand this possibility. Future work on this topic should also examine the relative effect, and the timing of the effects, of various disorders on social and workplace functioning.

Point 3: Line 61 is about homeless youth. The rest of the paragraph is about youth in general. Please clarify this so the reader knows when you speak of homeless youth versus youth overall.

Response 3: Thank you for your comment. We have updated the paragraph to read:

Among youth in general, the peak occurrence of most mental illnesses begins before the age of 25 [13,14] (Lines 62-63)

Point 4: Lines 80 and 81 also deserve a re-write, as do lines 90 to 92.

Thank you for this important paper.

Response 4: Thank you for your comments. Lines (81-82) [former lines 80-81] have been improved for clarity and conciseness and now read:

“Moreover, the specific mental illness experienced by youth could be important in predicting youth-onset homelessness”

Lines (91-93) [former lines 90-92] reads as:

“Based on the paucity of literature on mental illness as a reason for youth homelessness, the purpose of this study was to evaluate the potential role of mental illness in homelessness onset in youth versus adulthood among homeless adults.”

Reviewer 3 Report

Thank you for the opportunity to review the manuscript, Mental Illness and Youth-Onset Homelessness: A Retrospective Study amongst Adults Experiencing Homelessness. The paper was a pleasure to read. It is well structured and has a logical flow. As a non-north American I have enjoyed reading about research from the other side of the globe, in an area we are also grappling with in Australia, as other developed nations are too. I commend the authors on the paper and can see it is part of a suite of papers they have written.

Importantly, the paper provides a useful addition to the small evidence base around youth, mental illness and homelessness. The authors provide a number of practical applications of the findings, including the clear need for more early intervention focussed programs to ensure that young people with mental illness receive appropriate support to assist with their illness and potentially avert a homelessness pathway. They also rightly point to the need for young people experiencing homelessness also receive support with their mental health issues, as the combination of these ‘challenges’ is detrimental to wellbeing. I often say, like many homeless-serving practitioner colleagues, that if you were homeless and you didn’t have a mental health or addiction problem before that, you are very likely to after the homelessness experience. This message resonates with people and this paper provides some data to prove it!

Some minor comments for consideration if I may:

It would be helpful to understand when ‘youth’ starts. In Australia we tend to use two definitions: 15-24 and 12-24. I understand you are talking 18-24 in some of the paper, but is there a factor of things being seen as youth homelessness when actually they might be child homelessness? i.e. under 12 and therefore possibly homeless as part of a family unit.

At line 26 remover etc after violence, trauma. To me this suggests a level of insignificance.

Line 95, extra of in the sentence needs removing.

Table 1 is great. My only comment is: you add the variable insured/uninsured. Can you perhaps add a sentence in the discussion somewhere about what this means for people. Remember your audience is broad and international. Is it the case, as I assume, that uninsured means NO access or very limited access to MH support services?, i.e. accessibility is a challenge when no insurance? This is an area where there are clear opportunities to target support/solutions in early intervention.

Please check the journal convention re use of dashes, i.e. lines 216 and 218. I don’t know the convention, but what is used are hyphens and no en or em dashes for example.

I would be interested in a more cultural background analysis. Not for this paper. But in another one. So does cultural background and the other background variables (education etc.) correlate with higher risks of both MHI and youth-onset homelessness?

I couldn’t see too many typos etc, but there are a couple, so it might warrant someone doing a final pass over it with a keen proofing/editing eye.

Author Response

Response to Reviewer 3 Comments

Thank you for the opportunity to review the manuscript, Mental Illness and Youth-Onset Homelessness: A Retrospective Study amongst Adults Experiencing Homelessness. The paper was a pleasure to read. It is well structured and has a logical flow. As a non-north American I have enjoyed reading about research from the other side of the globe, in an area we are also grappling with in Australia, as other developed nations are too. I commend the authors on the paper and can see it is part of a suite of papers they have written.

Importantly, the paper provides a useful addition to the small evidence base around youth, mental illness and homelessness. The authors provide a number of practical applications of the findings, including the clear need for more early intervention focussed programs to ensure that young people with mental illness receive appropriate support to assist with their illness and potentially avert a homelessness pathway. They also rightly point to the need for young people experiencing homelessness also receive support with their mental health issues, as the combination of these ‘challenges’ is detrimental to wellbeing. I often say, like many homeless-serving practitioner colleagues, that if you were homeless and you didn’t have a mental health or addiction problem before that, you are very likely to after the homelessness experience. This message resonates with people and this paper provides some data to prove it!

Some minor comments for consideration if I may:

Point 1: It would be helpful to understand when ‘youth’ starts. In Australia we tend to use two definitions: 15-24 and 12-24. I understand you are talking 18-24 in some of the paper, but is there a factor of things being seen as youth homelessness when actually they might be child homelessness? i.e. under 12 and therefore possibly homeless as part of a family unit.

Response 1: Thank you so much for your comments regarding our manuscript. Youth homelessness comprises individuals less than 25 years of age captured in these two categories: <18years and >18years. The former represents children and usually goes from ages 13-17 years, whereas the latter represents “youth” from ages 18-24 years as utilized in the manuscript. There could be a factor(s) of things being seen in homeless youth that might be prevalent among homeless children, however, studies have shown the peak occurrence of most mental illnesses to be before 25, particularly in the youth category i.e. ages 18-24 years. Our sample only consists of individuals from age 18 years and above, hence, the emphasis of our work is on ages 18-24 years (as “youth”). However, we recognize that some of our literature is more general in nature and thus language could be expended to be more precise. In the revised manuscript, we have added the following for greater clarity (lines 41-43):

Homeless children and youth are described as individuals less than 25 years of age living in poverty and without a stable, reliable and adequate nighttime residence, and who may not have an alternative safe living environment [4,5].

And (lines 44-45):

Included in this group are accompanied and unaccompanied children (defined here as <18 years of age) and youth (defined here as 18-24 years of age) with or without a legal parent or guardian, respectively [6].

And (lines 47-48):

Also, most unaccompanied individuals experiencing homelessness (89%) are between the ages of 18 and 24 (i.e., “youth”), and are particularly vulnerable to the detrimental and negative effects associated with homelessness [6,7].

Point 2: At line 26 remover etc after violence, trauma. To me this suggests a level of insignificance.

Response 2: Thank you for bringing this to our attention. We have removed “etc” from line (67).

Point 3: Line 95, extra of in the sentence needs removing.

Response 3: Thank you for bringing this to our attention. We have removed the extra “of” from the sentence (line 94).

Point 4: Table 1 is great. My only comment is: you add the variable insured/uninsured. Can you perhaps add a sentence in the discussion somewhere about what this means for people. Remember your audience is broad and international. Is it the case, as I assume, that uninsured means NO access or very limited access to MH support services?, i.e. accessibility is a challenge when no insurance? This is an area where there are clear opportunities to target support/solutions in early intervention.

Response 4: Thank you for bringing this to our attention. In our study, uninsured means an endorsement of “I do not have health insurance” whereas insured means “I have some type of health insurance.” Although insurance status is not a perfect proxy for health care access, studies have indeed shown accessibility to healthcare as a challenge for those without insurance or limited insurance.

To address the Reviewer’s comments, we have added to the revised discussion (lines 312-314):

“It is important to note that lack of insurance, however, could pose as a barrier to the accessibility of mental health and other support services, whereby access is limited for youth who are uninsured [68]”

Reference #68:

Cummings, J. R.; Wen, H.; Druss, B. G. Improving access to mental health services for youth in the United States. JAMA. 2013, 309(6), 553–554. https://doi.org/10.1001/jama.2013.437.

We have also added (lines 146-149):

“Participants were asked about their education (<GED/high school vs >GED/high school), race (white, black, vs other), sex (male vs female), insurance status (insured vs uninsured), employment status (employed vs unemployed), marital status (married vs unmarried), and annual income (<$5,000 vs >$5,000)”

Point 5: Please check the journal convention re use of dashes, i.e. lines 216 and 218. I don’t know the convention, but what is used are hyphens and no en or em dashes for example.

Response 5: Thank you for bringing this to our attention. We have updated the pertinent sections with hyphens. (lines 239 and 240).

Point 6: I would be interested in a more cultural background analysis. Not for this paper. But in another one. So does cultural background and the other background variables (education etc.) correlate with higher risks of both MHI and youth-onset homelessness?

Response 6: Thank you for this suggestion – we will absolutely consider this examination for a future manuscript! Please note that in our revision, we now add results of the significant covariates in our main analyses (lines 191-195), which at least partially addresses this suggestion:

“Additionally, being other race (vs white; p=0.0202) and having fewer months of homelessness in the current homelessness episode (p=0.0178), more homelessness episodes over the lifetime (p<0.0001), and being from the Dallas study site (p=0.0303) were associated with greater odds of youth- versus adult-onset homelessness in this analysis”

And lines (197-215)

Participants who were diagnosed with bipolar disorder and schizophrenia/schizoaffective disorder had 1.746 (95% CI: 1.242 – 2.453) and 1.831 (95% CI: 1.223 – 2.739) times greater odds of experiencing youth- versus adult-onset homelessness than those with no bipolar disorder or schizophrenia/schizoaffective disorder diagnosis, respectively. Moreover, being other race (vs white; p=0.0224) and having fewer months of homelessness in the current homelessness episode (p=0.0182), and more homelessness episodes over the lifetime (p<0.0001) were associated with greater odds of youth- versus adult-onset homelessness in analysis with bipolar disorder. Fewer months of homelessness in the current homelessness episode (p=0.0108), more homelessness episodes over the lifetime (p<0.0001), and being from the Dallas study site (p=0.0278) were associated with greater odds of youth- versus adult-onset homelessness in analysis with schizophrenia/schizoaffective disorder. However, the association between having major depression and homelessness onset was not significant (p=0.1040). Having fewer months of homelessness in the current homelessness episode (p=0.0172) and more homelessness episodes over the lifetime (p<0.0001), and being from the Dallas study site (p=0.0264), however, were associated with greater odds of youth- versus adult-onset homelessness in this analysis. Lastly, the association between having anxiety disorder, PTSD and homelessness onset on a subsample (Study 2) was not significant in covariate-adjusted analyses (p=0.4841 and p=0.6789, respectively). However, having more homelessness episodes over the lifetime (p<0.0001) was associated with greater odds of youth- versus adult-onset homelessness in analyses with other anxiety disorders and PTSD.

And lines (219-222)

“Additionally, being other race (vs white; p=0.0213) and having fewer months of homelessness in the current homelessness episode (p=0.0153), more homelessness episodes over the lifetime (p<0.0001), and being from the Dallas study site (p=0.0275) were associated with greater odds of youth- versus adult-onset homelessness.”

And lines (226-229)

“Additionally, being other race (vs white; p=0.0207), and having fewer months of homelessness in the current homelessness episode (p=0.0122), more homelessness episodes over the lifetime (p<0.0001), and being from the Dallas study site (p=0.0486) were associated with greater odds of youth- versus adult-onset homelessness in this analysis.”

Point 7: I couldn’t see too many typos etc, but there are a couple, so it might warrant someone doing a final pass over it with a keen proofing/editing eye.

Response 7: Thank you for bringing this to our attention. We have read the paper in its entirety and ensured no typos were present.
